# The influence of roads on the fine-scale population genetic structure of the dengue vector *Aedes aegypti* (Linnaeus)

**Maria Angenica F. Regilme**[1,2,3,4], **Thaddeus M. Carvajal**[1,2,3,4], **Ann–Christin Honnen**[5,6], **Divina M. Amalin**[3,4], **Kozo Watanabe**[1,2,3,4]*

**1** Center for Marine Environmental Studies (CMES), Ehime University, Matsuyama, Japan, **2** Graduate School of Science and Engineering, Ehime University, Matsuyama, Japan, **3** Biological Control Research Unit, Center for Natural Science and Environmental Research, De La Salle University, Manila, Philippines, **4** Department of Biology, De La Salle University, Manila, Philippines, **5** Swiss Tropical and Public Health Institute, Basel, Switzerland, **6** University of Basel, Basel, Switzerland

* watanabe.kozo.mj@ehime-u.ac.jp

**Data Availability Statement:** The dataset was made publicly available and deposited at www.vectorbase.org with the population biology project

## Abstract

Dengue is endemic in tropical and subtropical countries and is transmitted mainly by *Aedes aegypti*. Mosquito movement can be affected by human-made structures such as roads that can act as a barrier. Roads can influence the population genetic structure of *Ae. aegypti*. We investigated the genetic structure and gene flow of *Ae. aegypti* as influenced by a primary road, España Boulevard (EB) with 2000-meter-long stretch and 24-meters-wide in a very fine spatial scale. We hypothesized that *Ae. aegypti* populations separated by EB will be different due to the limited gene flow as caused by the barrier effect of the road. A total of 359 adults and 17 larvae *Ae. aegypti* were collected from June to September 2017 in 13 sites across EB. North (N1-N8) and South (S1-S5) comprised of 211 and 165 individuals, respectively. All mosquitoes were genotyped at 11 microsatellite loci. AMOVA $F_{ST}$ indicated significant genetic differentiation across the road. The constructed UPGMA dendrogram found 3 genetic groups revealing the clear separation between North and South sites across the road. On the other hand, Bayesian cluster analysis showed four genetic clusters (K = 4) wherein each individual samples have no distinct genetic cluster thus genetic admixture. Our results suggest that human-made landscape features such as primary roads are potential barriers to mosquito movement thereby limiting its gene flow across the road. This information is valuable in designing an effective mosquito control program in a very fine spatial scale.

## Author summary

Dengue, a mosquito-borne viral infection is a serious health problem in tropical and subtropical countries such as Philippines. Most dengue prevention programs aim to eradicate its mosquito vector, *Aedes aegypti*. A successful population control program is reliant in understanding the mosquito behavior and ecology including how human-made structures

ID VBP0000716 and link at https://vectorbase.org/popbio-map/web/?projectID=VBP0000716.

**Funding:** This study was supported in part by the Japan Society for the Promotion of Science (JSPS) Grant-in-Aid Fund for the Promotion of Joint International Research (Fostering Joint International Research (B)) under grant number 19KK0107, JSPS Grant-in-Aid for Scientific Research (A) under grant number 19H01144, the JSPS Core-to-Core Program B. Asia-Africa Science Platforms, and the Endowed Chair Program of the Sumitomo Electric Industries Group Corporate Social Responsibility Foundation. The funders had no role in study design, data collections and analysis, decision to publish, or preparation of the manuscript.

**Competing interests:** The authors have declared that no competing interests exist.

such as roads influence its expansion and movement. Previous studies have discovered the barrier effect of roads in the movement of mosquitoes. In this study, we examined the influence of roads in the population genetic structure of *Ae. aegypti* in a fine spatial scale using 11 microsatellite markers. We found significant genetic differentiation of mosquito populations across the road. Our results suggest limited gene flow across the road and supports our hypothesis that roads are potential barriers to mosquito dispersal. This information can be used in designing an effective mosquito population control zones in perceived barrier to mosquito dispersal such as roads.

## Introduction

Dengue is an infectious disease transmitted by *Aedes* mosquitoes. The main dengue vector, *Aedes aegypti* is best adapted to urban environments. Dengue prevention programs are usually focusing on eliminating mosquito larval habitats and adult mosquito populations [1,2]. Successful dengue control programs are dependent upon understanding the population genetic structure of *Ae. aegypti* including how human-made structures such as roads influence its dispersal. For example, estimating gene flow and barriers to dispersal such as roads can help in predicting the spread of insecticide resistance genes in *Ae. aegypti* populations [3] and *Wolbachia*-infected mosquito populations release [4,5].

Most population genetic studies of *Ae. aegypti* were usually described at a country spatial scale [6–8]. In contrast, fine spatial scale genetic analysis (e.g scale of several households or city block) though challenging is still feasible as evidenced in several studies that revealed significant genetic differentiation of *Ae. aegypti* at fine spatial scales. For example, significant genetic differentiation was found at spatial scales of 5 km to 2,000 km using ND4 mitochondrial gene [9] and microsatellite markers [10–12]. Recently, Carvajal et al [13,14] revealed the low genetic differentiation and high gene flow among *Ae. aegypti* populations in Metropolitan Manila, Philippines, which suggest the influence of passive and active dispersals of the mosquitoes to population genetic structure. Dispersal ability is a determinant factor of population genetic structure and the genetic effects of habitat fragmentation at fine spatial scales [15]. Population genetics can be utilized in estimating dispersal ability through spatial autocorrelation analysis. For example, limited spatial ranges of significant spatial autocorrelations of up to 1 km suggested the active dispersal capability of *Ae. aegypti* at microgeographic areas of eastern Thailand [12] and in Metro Manila, Philippines [13].

Although the genetic effect of roads in the dispersal pattern of *Ae. aegypti* in a fine spatial scale is very limited, the influence of roads was often studied through conventional mark-release-recapture method (MRR). This method is labor-intensive and the rearing and marking procedure can affect the mosquito fitness and movement in the field [16]. Studies in MRR of *Ae. aegypti* demonstrated that the dispersal of this mosquito vector could be influenced by the type of road [17–18]. *Ae. aegypti* prefer crossing smaller and quieter roads as compared to larger and busy roads [17–18]. To date, the only study that investigated the effect of road such as highway on *Ae. aegypti* using genetic approach was done by [19]. Using larval samples, [19] found significant genetic differentiation across a 900-meter-long stretch and 120-meters-wide highway. Our study, provided an extensive sampling of adult mosquitoes and larval survey in a primary road, España Boulevard (EB). As compared to highways, primary roads are characterized with presence of several households, buildings and minor residential roads across that could serve as a pathway for public and private transportation to traverse. We expect that

passive method of mosquito dispersal in primary roads are more likely possible than in highways because of the presence of minor roads that can aid the movement across.

Furthermore, genetic analysis may also reveal co-occurrences of multiple genetic clusters at fine-spatial scale and their genetic admixture in mosquito individuals. For example, previous studies of *Ae. aegypti* revealed sympatric numerous genetic clusters (K = 3 to K = 16) from 30 km up to 2, 000 km [10,11,13,20], which might be due to the divergence from a single ancestry resulting into multiple genetic clusters over time and the random distribution of *Ae. aegypti* populations from nearby cities, regions or country. A recent study in the region of Metropolitan Manila, Philippines [13], discovered the probable number of genetic clusters of K = 4 in a fine spatial scale in Metro Manila, Philippines and genetic admixture in *Ae. aegypti* individuals. In this study, we narrow down the spatial scale up to 2 km to test if we could still observe multiple genetic clusters. This information is important prior to analysis of population genetic structure and gene flow because it can give background information on how the variety of genes are co-existing in a limited spatial scale.

Here, we studied the influence of road on the population genetic structure and gene flow of *Ae. aegypti* using 11 microsatellite loci, to analyze the genetic relatedness among the mosquito populations and to determine if multiple genetic clusters in a very fine spatial scale can still be observed. We hypothesize that *Ae. aegypti* populations across a 2000-meter-long stretch and 24-meters-wide road may have differences in its population genetic structure because it acts as a potential barrier in mosquito movement.

## Methods

### Study site

The study selected a certain area in City of Manila, Philippines. This area consists of two traversing roads, a primary road España Boulevard (EB) and a secondary road AH Lacson Avenue (LA). EB is the selected study area and is divided into North and South sides. It is a primary road based on Philippine geographic information system (PhilGIS). It is located within a highly urbanized area in Metro Manila consisting of commercial, residential, and industrial infrastructures and it connects two cities: Manila City and Quezon City. EB as a primary road have 44 intersections connecting two or more roads across the north and south. It is one of the busiest roads in the city of Manila with heavy traffic congestion and high human population density. Some shaded areas such as trees can be found across EB. The mean width of the road (EB) sampled is 24.27 meters and its length is 2, 000 meters with coordinates of 14˚ 37′ 3″ N, 121˚ 0′ 4″ E.

### Collection, sampling and identification

We used a two stage cluster systematic sampling design to randomly select households for collecting mosquitoes. We used OpenEPI software [21] to calculate the target sample size. Equal allocation and a design effect of 3 [22,23] was used to calculate the target sample size. The estimate of *p* used in the calculation was 0.23 according to the study of [24]. The alpha level was set at 95% (α = 1.96). The maximum tolerable error was equal to 10%. We computed the sample size on OpenEpi online software (https://www.openepi.com/Menu/OE_Menu.htm) [21]. An additional 15% allowance was added to provide a buffer, including refusal to participate, yielding a target sample size of 236 households per stratum (North and South; *n* = 472). Each sampling site (n = 13) is comprised of 1 to 5 smallest administrative division within the city (barangay; n = 35) and is defined as the population of this study. Households (n = 7 to 25 per sampling site) were selected based on their voluntary informed consent for mosquito collection and larval survey (Fig 1). The sampling unit of our study was the household defined as

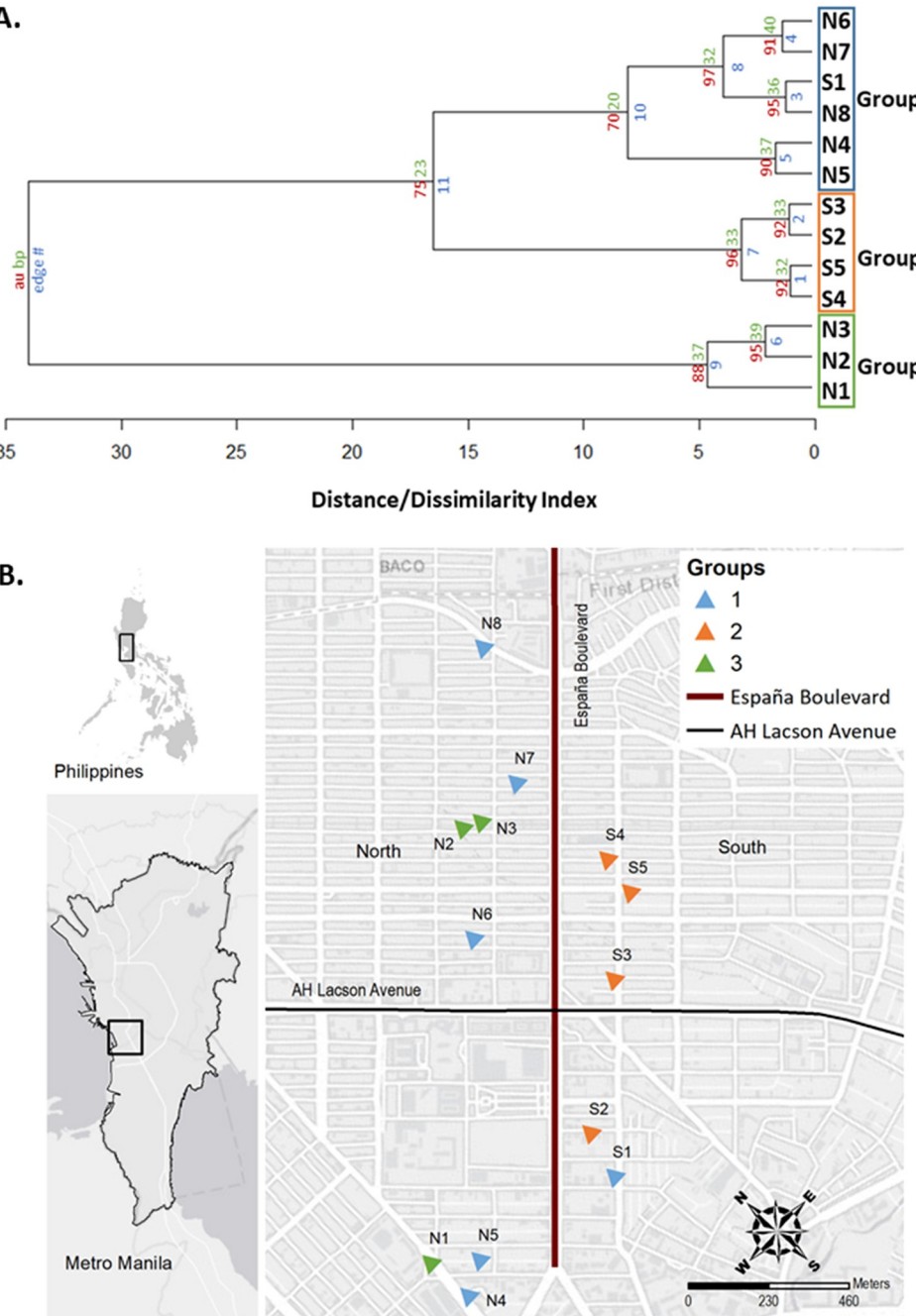

**Fig 1. (A) An unweighted pair group method with the arithmetic mean (UPGMA) dendrogram and (B) map of the 13 sites where the *Ae. aegypti* were collected in households across EB highlighted in red and in black is LA, an important East-west connection.** For more detailed information, please see S1 Table. The map was created using ArcGIS version 10.2.2 from http://landsat.gsfc.nasa.gov/images/. **A.** A dendrogram based on the unweighted pair group method with the arithmetic mean (UPGMA) using the pairwise genetic distance among the 13 sites from the north and south of EB road. The red numbers indicate the approximately unbiased (AU) p-values [44–45] and the green numbers are the standard bootstrap value (BP) [46] using the r package pvclust [47]. **B.** Map showing the distribution of the genetic groups as observed in the UPGMA cluster dendrogram (Fig 2A). The color of the triangles corresponds to the results of the dendrogram.

one unit of accommodation [25]. The geographic coordinates of the sampled households were recorded. In order to obtain a single geographic coordinate for each sampling site, we calculated the geographic midpoint [26] of sampled households in their respective sites.

To provide more information about the effect of roads, we collected adult mosquitoes and representative larval samples from each water holding containers across a primary road with presence of several smaller roads that can potentially carry mosquito migrants across. The sampling method of collection of both adult *Ae. aegypti* and larvae though labor intensive is more informative as compared to only larval samples. Collections of adult and larval mosquitoes in each household were performed simultaneously in the different sites during the rainy season and high dengue cases, from June to September 2017. Adult mosquitoes were collected by installing commercially available mosquito UV light trap (MosquitoTrap, Jocanima Corporation, Las Piñas, Philippines) for 48 hours either inside or outside of each household. The UV light trap produces heat and $CO_2$ gas attracting mosquitoes to enter into the capture net via a strong current from the ventilator [27]. In each household, one mosquito trap was installed. We also surveyed potential water breeding sites on each of the household premises and collected larvae when present. Mosquitoes and larvae collected were identified morphologically to species level using the keys for adult and immature mosquitoes of Rueda et al [28]. All collected samples were preserved in RNALater and stored at -20˚C prior to nucleic acid extraction.

We analyzed 211 *Ae. aegypti* from 106 households in the North area and 165 *Ae. aegypti* collected from 104 households in the South area of the road (S1 Table). The number of *Ae. aegypti* collected per site ranged from 20 to 46 individuals.

## DNA extraction, PCR amplification and microsatellite genotyping

Simultaneous DNA and RNA extraction in individual mosquito adult and larval samples (*n = 376*) was performed using the Qiagen AllPrep DNA/RNA micro kit following the manufacturer's protocol with some modifications. In this study, the extracted DNA was only used while the extracted RNA were kept in– 80˚C for future investigations (e.g. transcriptome analysis). DNA quality was checked in NanoDrop 2000 Spectrophotometer (Thermo Scientific). We used the same 11 microsatellite primer and PCR protocol that have been used in an earlier study by Carvajal et al [13]. All PCR amplifications were performed with 1μl of genomic DNA in a final volume of 10 μl. We performed 4 sets of multiplex PCRs with 3 pairs of loci grouped together (S2 Table). All PCR amplifications were composed of the following: 10x Ex Taq buffer, 25mM $MgCl_2$, 2.5mM dNTP, 5% dimethyl sulfoxide (DMSO), 10μm of fluorescently labelled forward primer, 10μm of reverse primer, and 5 units/μl of Takara Ex Taq (Takara Bio Inc.). Each multiplex PCR amplification was conducted as follows: an initial denaturation of 94˚C, denaturation at 94˚C, annealing varies in each set of multiplex PCR (57˚C to 62˚C), extension at 72˚C, and final extension of 72˚C. PCR products were analyzed in 3% agarose gel electrophoresis stained with Midori Green Advance DNA stain.

Multiplex PCR products were diluted in 1:3 water and pooled into final volume of 16 μl. Samples were prepared prior to fragment analysis with 1 μl of pooled PCR product added with 0.5 μl GeneScan 600 LIZ dye standard and 10 μl HI-DI Formamide. Fragment analysis was performed using SeqStudio Genetic Analyzer (Applied Biosystems). We used PeakScanner (ThermoFisher Scientific) to identify peak and fragment size and Microsatellite Analysis app (ThermoFisher Scientific) for genotyping.

## Data analysis

Allele scores were checked for genotyping errors and for the presence of null alleles using Microchecker [29]. The observed heterozygosity ($H_o$), expected heterozygosity ($H_e$), mean

number of alleles, mean number of effective alleles, allelic richness, mean number of allele frequency and mean number of private alleles were computed in GenAlEx version 6.51b2 [30]. We calculated the Inbreeding coefficient ($F_{IS}$) for all loci across populations following [31] and tested statistical deviation from Hardy-Weinberg equilibrium (HWE) using Genepop web version [32]. The markov chain parameters were set at 10,000 dememorizations, 100 batches and 5,000 iterations for testing deviations from HWE.

To test the statistical significance of genetic variations among groups (North and South of EB), among sites (N1 to N8 and S1 to S5) within groups, and within sites (N1 to N8; S1 to S5), we computed the Analysis of Molecular Variance (AMOVA) using Arlequin version 3.5.2.2 [33] with 10,000 permutations. We assessed the degree of genetic differentiation between the 13 sampling sites by calculation of the pairwise $F_{ST}$ values in Arlequin. In order to determine if the mean values of the pairwise $F_{ST}$ within groups (North versus South) and between groups are significantly different from each other, we performed Mann- Whitney U-test.

Dendrograms among the sites were constructed using the genetic distance matrix (pairwise $F_{ST}$ values) generated from Arlequin software. We employed the Unweighted Pair Group Method with the arithmetic mean (UPGMA) method using the APE package [34] and R program [35]. To determine the optimal number of groups in the dendrogram, we used the pseudo-$t^2$ index from the package NbClust [36] of R program.

To infer the individual assignment of *Ae. aegypti* to genetic clusters whose members share similar genetic characters, we used the Bayesian clustering algorithm in STRUCTURE version 2.3.4 software [37]. We used the same parameter set as in [13] testing for 1–20 presumed genetic clusters (K) with 20 iterations per K, a burn-in period of 200,000 steps and 600,000 Markov Chain Monte-Carlo (MCMC) replications using an admixture model with correlated allele frequencies. The best estimate of K was calculated with the ad-hoc statistic Δ*K* as described by [38] using Structure Harvester Web version 0.6.94 (http://taylor0.biology.ucla.edu/structureHarvester/#) [39]. We visualized the final barplots using the R package *pophelper* [40] as implemented in R program.

To estimate the migration rate between the North and South sites and within North and South, we used the GENECLASS v2.0 [41]. For each individual *Ae. aegypti* in a population, we determined the probability that it belongs to its home population, probability of being a migrant across the road and the probability of being a migrant within North or South. We used the Bayesian criterion of [42] to identify the first generation migrants and the likelihood computation of Lhome/Lmax with a Monte-Carlo resampling algorithm [43]. We used the following parameter sets: 10,000 simulations and a threshold probability value of 0.05. In order to determine if the proportion of the migration rate across the road is lower than that within North or South, we performed Z-score for 2 populations proportions.

The test for Isolation by distance was performed using a Mantel's test in GenAlex to determine if geographical distance influence the genetic differentiation. The pairwise genetic distance ($F_{ST}$) was compared to the geographical distance (km) among the sites. To obtain the geographic distances between sites we used the geographic midpoint of the sampled households per site calculated based on the coordinates (latitude and longitude) of the households. All Mantel tests were assessed for the significance of the correlation using permutation tests (9999 permutations).

To further evaluate whether genetic variation was correlated with geographic distance, we performed a spatial autocorrelation analysis using GenAlEx [30]. We computed the autocorrelation coefficient (*r*) from the geographic distance as described above and the genetic distance (pairwise $F_{ST}$ values). This measure determines the genetic similarity between the 13 sites within an identified geographic distance class. We identified the suitable distance class based

on the observed distribution of pairwise geographic distance between sites. We used 14 distance classes at 0.10 km interval.

## Results

### Genetic diversity and differentiation

The mean number of alleles (MNa) per sampling site ranged from 7.73 (N6) to 12.73 (N1) while the mean number of effective alleles (Mne) ranged from 3.28 (S5) to 4.58 (N1) (S1 Table). In contrast, the mean number of allele frequency ranged from 3.64 (S4 and S5) to 4.91 (N6) between sites and the mean number of private alleles ranged from 0.00 (N6) to 1.6 (N1) between sites. All 13 sites displayed significant non-conformance to Hardy-Weinberg equilibrium (He > Ho) after Bonferroni correction which implies heterozygosity deficiency that can be caused by inbreeding, the expected heterozygosity (He) ranged from 0.60 (S5) to 0.72 (N8).

AMOVA results showed significant genetic differentiation ($F_{ST}$ = 0.0268) between sampling sites North (N1 –N8) and South (S1 –S5) of the EB road (Table 1). Small but significant estimates among ($F_{SC}$) and within ($F_{CT}$) 13 sites (N1 to N8; S1 to S5) were observed. Population pairwise $F_{ST}$ between the combined all northern and all southern sites showed significant genetic differentiation ($F_{ST}$ = 0.0321). A significant difference between the mean pairwise $F_{ST}$ within groups (North and South; mean = 0.0321) and between groups (mean = 0.0337) were found using the Mann-Whitney *U*-test at P < 0.05. The pairwise $F_{ST}$ among the 13 sites ranged from 0.0024 (N1 and N5) to 0.0818 (N4 and N6). Among these comparisons of pairs of sites, 53 out of 91 (58.24%) pairwise $F_{ST}$ values presented significant genetic differences (S3 Table).

The number of groups identified on the UPGMA Dendrogram was three based on the cindex index. The groupings (Fig 1) revealed the clear separation of North (Group 3– N1 to N3) and South (Group 2 –S2 to S5) of EB. On the other hand, genetic similarity between one site from the South and North of EB were shown for group 1 (N4 to N8 and S1).

### Genetic structure

In STRUCTURE analysis, the most probable number of genetically differentiated clusters across the mosquito populations was K = 4 (S1 Fig). The barplot (Fig 2) displays the distribution of the assumed genetic clusters of each *Ae. aegypti* in the North (N1 to N8) and South (S1 to S5) of the road. The barplot suggests admixture of the genetic clusters across all the mosquito individuals. STRUCTURE barplot at non-optimal k-values at K = 2,3,5,6 and 7 (S2 Fig) also displayed genetic admixture.

### Migration rate estimates between North and South sites and within North and South

We identified 94 individuals out of 376 (25%) as potential first-generation (F0) migrants (S4 Table), of which 26 (6.91%) were migrants across the road and 68 (18.08%) were migrants

**Table 1. Analysis of molecular variance (AMOVA) using a panel of 11 microsatellites.**

| Variation | ss | vc | pv | FI |
|---|---|---|---|---|
| Among North & South | 18.509 | 0.0149 | 0.2810 | $F_{ST}$ = 0.0268* |
| Among sites within North and South | 136.329 | 0.1274 | 2.3995 | $F_{SC}$ = 0.0241* |
| Within sites | 3725.183 | 5.1670 | 97.3196 | $F_{CT}$ = 0.0028* |

ss = sum of squares; vc = variance components; pv = percentage variation; F-statistics for each hierarchy; $F_{ST}$ = among groups; $F_{SC}$ = among populations within groups; $F_{CT}$ = within populations;

*P < 0.05

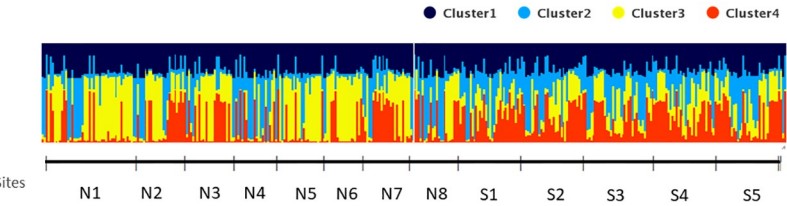

**Fig 2. STRUCTURE bar plot displaying the assignment probabilities of each genotyped *Ae. aegypti* individual grouping into 4 clusters.** Each individual is represented by a single vertical horizontal line. Brackets are shown to separate collection sites of North (N1 to N8) and South (S1 to S5).

within South or North. The migration rate across the road (6.91%) was significantly lower than that within North or South (18.08%) based on the z-score test for two populations proportions at p <0.05.

## Isolation by distance and spatial autocorrelation

The Mantel test analysis based on the pairwise genetic distances ($F_{ST}$) and the geographic distances between all populations was not significantly correlated (r = 0.05, p-value = 0.34) (S3 Fig) thus indicating no isolation by distance. Non-significant results were also obtained in the northern populations (r = 0.03, p-value = 0.39) and in the southern populations (r = 0.29, p-value = 0.11). On the other hand, the spatial autocorrelation also showed a negative correlation between genetic and geographic distance (S4 Fig).

## Discussion

Our findings support the hypothesis that human-made structure such as roads can create a barrier to dispersal of *Ae. aegypti* on a very fine spatial scale. The significant $F_{ST}$ estimate (0.0268) across the EB revealed genetic differentiation on both sides (North and South) of EB as supported by the significant population pairwise $F_{ST}$ ($F_{ST}$ = 0.0321) between the combined all North sites and all South sites. Our result is consistent with a study conducted in a 900-meter long and 120-meter wide road in West Indies, which found a small but significant $F_{ST}$ value (0.011 to 0.021) across the road using 9 microsatellite loci and 2 SNP markers [19].

The results of the cluster dendrogram showed the clear separation of North sites and South sites that further supported our hypothesis of barrier effect of roads in mosquito movement. Group 2 (Fig 1B, orange triangle) illustrates the clustering of the genetically similar *Ae. aegypti* populations from the south side of the EB while Group 3 (Fig 1B, green triangle) showed the clustering of populations from the north sites. The EB road may separate the two areas potentially limiting the migration between mosquito populations thereby resulting in the formation of genetic groups as seen in the results of the cluster dendrogram. In a previous study of Carvajal et al [13] in Metropolitan Manila, it was inferred that unique landscape features such as highways or rivers could potentially result to the formation of genetic groups. It was further discussed that such landscape feature may generally preserved the alleles within that mosquito population, thereby making it distinct from other populations overtime due to limited gene flow.

Our results of the cluster dendrogram and pairwise $F_{ST}$ are interesting because the mean width of the road is around 24.27 m and given the dispersal capability of *Ae. aegypti* ranging from 100 m to 800 m [48–52], we expected no distinct groupings and wide adult dispersal unless the mosquitoes were not able to successfully cross EB road. Despite the fact that the distance of the EB road is within the dispersal estimates of *Ae. aegypti*, possibly the inadequate

cover and shade from trees and vegetation across the EB road made it unsuitable for the mosquitoes to traverse across as previously explained by Hemme et al [19]. The degree of shade has strong interrelation with the presence of *Ae. aegypti* in mosquito and larval surveys [53]. Mosquitoes tend to disperse in areas with numerous water breeding containers and trees that can provide heavy shading while busy roads seem to inhibit the mosquito movement [18]. The lack of available oviposition sites and suitable blood meal hosts [19] are possibly some of the factors that have prevented *Ae. aegypti* movement across EB road.

Interestingly, Group 1 (N4 to N8 and S1) (Fig 1A, blue square) of the dendrogram displayed clustering of one site from the south and north. The co-presence of South sites (S1) and North sites (N4 to N8) in Group 1 could be attributed that a part of the EB is not a potential barrier and migration of mosquitoes could be possible. For example, the presence of secondary roads (e.g LA road and the intersections) probably facilitated the passive dispersal of mosquitoes between sites N4 to N7 and site SI for example through increased human-mediated dispersal of *Ae. aegypti*. The presence of minor roads such as residential, pedestrian lane and intersections can act as a route of passive mosquito dispersal. Minor roads could be a possible route for mosquitoes to traverse the road. Small pedestrian lanes provide passive mosquito dispersal by humans. For example, when mosquitoes are accidentally transported by land vehicles that can be over the flight distance capability of a typical *Ae. aegypti* [52,54,55]. Future investigations are necessary to validate the kinds of conclusions that can be drawn from the effects of ecological factors mentioned previously. Alternative mechanisms that drive the mixed structure in group 1 might be because of common source population/s between the sites in group 1 that allow to share same alleles across the road.

We detected multiple genetic clusters (K = 4) in a very fine spatial area by Bayesian analysis, which was concordant with the results (K = 3 to 4) from a previous study of *Ae. aegypti* among 11 sites not more than 30 km apart in Sao Paulo, Brazil [11]. Previous population genetic studies of *Ae. aegypti* from Philippines also displayed multiple genetic clusters (K = 2 to K = 6) in fine-spatial scale using microsatellite markers [13,56,57]. The different ancestry populations of our samples may be the reason of the co-occurrence of the four genetic clusters in the small area rather than the limited gene flow in the study area. Despite the occurrence of multiple genetic clusters (K = 4), our results revealed genetic admixture thus no distinct genetic cluster observed. The genetic admixture might indicate that the individual *Ae. aegypti* from these sites could potentially share alleles possibly due to the several mosquito invasions from neighboring cities surrounding the study area as observed in Philippines [13], China [58] and in the USA [59].

Our sampling strategy of collecting adult and representative larval samples from surveyed water breeding containers though challenging is more informative in the analysis of mosquito populations in a very fine spatial scale. We assume that our sampling strategy of adult mosquitoes' collection are more likely to represent the sampling site as compared to collecting or sampling eggs or larvae from the same water container [60]. Adult mosquito sampling within household can increase the chance of getting higher genetic variability as compared to only larval sampling. In contrast to [19], our study sampled adult mosquitoes (n = 359) and single larva from each surveyed water breeding container (n = 17) to minimize the possibility of sampling family members as seen from the previous studies of [61,62,63,64]. The collection of one larvae per water breeding container throughout the sampling site minimizes the probability of larvae from the same progeny [65].

Overall, the findings of this study displayed strong evidence of limited gene flow across the highway causing habitat fragmentation of the mosquito populations from the north and south of EB road. The results suggest that human-made structures such as primary road are potential barriers to mosquito dispersal limiting its movement across the road. Understanding the

dispersal pattern of *Ae. aegypti* in a very fine-spatial scale can give insights in predicting the spread of dengue virus infection. This information can also be used in the design of successful vector control strategies such as mosquito elimination programs in a very fine spatial scale. For example, we can use the information on the blocking potential of roads in *Wolbachia*-infected mosquito release programs. Road can be used as a unit of release as compared to city-wide mosquito release. Local elimination of *Ae. aegypti* can also be achieved by assigning control zones along roads that can potentially block the mosquito movement. Road blocking information can be used during dengue outbreaks wherein vector control agencies can determine high risk areas in the control zones. Knowledge on the effect of human-made structures such as roads in mosquito dispersal can greatly improve the implementation of a successful mosquito control programs [19].

## Supporting information

**S1 Fig. Identified optimal number of clusters (K) as calculated in the Structure Harvester program.**
(TIF)

**S2 Fig. STRUCTURE bar plot at non-optimal k-values (K = 2, 3, 5, 6 and 7).**
(TIF)

**S3 Fig. Mantel test on the relationship between genetic distance (pairwise $F_{ST}$) and geographic distance distance between to sites in km of the 13 sites of *A. aegypti*.**
(TIF)

**S4 Fig. Results of the spatial autocorrelation analysis as the influence of geographic distance on the genetic distance.**
(TIF)

**S1 Table. Summary of variation at 11 microsatellites, north and south of the EB road.**
(XLSX)

**S2 Table. List and characteristics of microsatellite markers used in genotyping.**
(XLSX)

**S3 Table. Pairwise $F_{ST}$ values in bold indicates significant p value $<0.05$. Sites are labelled as North (N1 to N8) and South (S1 to S5).**
(XLSX)

**S4 Table. Summary of the first generation migrants in the North (N1 to N8) and South (S1 to S5) populations.** The individuals classified as potential migrants ($P < 0.01$) are indicated in red, and the most likely population in green. Highlighted in yellow are the migrants across the road.
(XLSX)

## Acknowledgments

We are thankful to all the households that participated in this study for giving us their consent to collect mosquitoes and to the village officials of Sampaloc, Manila for all their help and support during the collection. We are grateful to Johanna Beulah Sornillo of Research Institute for Tropical Medicine, Philippines for her technical expertise and assistance in the statistical sampling design of our study. We would also like to thank Dr. Crystal Amiel Estrada of the University of the Philippines–Manila for providing her valuable comments regarding the

conceptualization of the research study. We would like to thank Katherine Viacrusis and Tatsuya Inukai for their technical assistance during the mosquito collection, Dewi Gustari for her assistance in the PCR experiments and Micanaldo Francisco for his technical assistance in constructing the sampling maps of this study. The authors are also thankful to Dr. Mary Jane Flores for her suggestions about the sampling strategy. We are also thankful to Dr. Maribet Gamboa, Dr. Michael Monaghan and Joeselle Serrana for their valuable suggestions on the population genetic analyses.

## Author Contributions

**Conceptualization:** Maria Angenica F. Regilme, Kozo Watanabe.

**Data curation:** Maria Angenica F. Regilme, Thaddeus M. Carvajal, Divina M. Amalin.

**Formal analysis:** Maria Angenica F. Regilme, Thaddeus M. Carvajal, Ann–Christin Honnen, Divina M. Amalin, Kozo Watanabe.

**Funding acquisition:** Kozo Watanabe.

**Investigation:** Maria Angenica F. Regilme, Thaddeus M. Carvajal, Divina M. Amalin.

**Methodology:** Maria Angenica F. Regilme, Thaddeus M. Carvajal, Divina M. Amalin, Kozo Watanabe.

**Project administration:** Kozo Watanabe.

**Resources:** Kozo Watanabe.

**Supervision:** Maria Angenica F. Regilme, Kozo Watanabe.

**Validation:** Maria Angenica F. Regilme, Thaddeus M. Carvajal, Ann–Christin Honnen, Divina M. Amalin, Kozo Watanabe.

**Visualization:** Maria Angenica F. Regilme, Thaddeus M. Carvajal, Ann–Christin Honnen, Divina M. Amalin, Kozo Watanabe.

**Writing – original draft:** Maria Angenica F. Regilme.

**Writing – review & editing:** Maria Angenica F. Regilme, Thaddeus M. Carvajal, Ann–Christin Honnen, Divina M. Amalin, Kozo Watanabe.

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
