## [Decision Letter · Decision Letter 0]

15 Oct 2020

Dear Dr. Watanabe,

Thank you very much for submitting your manuscript "The influence of roads on the fine-scale population genetic structure of the dengue vector Aedes aegypti (Linnaeus)" for consideration at PLOS Neglected Tropical Diseases. As with all papers reviewed by the journal, your manuscript was reviewed by members of the editorial board and by several independent reviewers. The reviewers appreciated the attention to an important topic. Based on the reviews, we are likely to accept this manuscript for publication, providing that you modify the manuscript according to the review recommendations. 

Sincerely,

Mariangela Bonizzoni

Associate Editor

Francis Jiggins

Deputy Editor

Reviewer's Responses to Questions

**Key Review Criteria Required for Acceptance?**

**Methods**

-Are the objectives of the study clearly articulated with a clear testable hypothesis stated?

-Is the study design appropriate to address the stated objectives?

-Is the population clearly described and appropriate for the hypothesis being tested?

-Is the sample size sufficient to ensure adequate power to address the hypothesis being tested?

-Were correct statistical analysis used to support conclusions?

-Are there concerns about ethical or regulatory requirements being met?

Reviewer #1: yes to all 

no ethical concern

Reviewer #2: (No Response)

Reviewer #3: Overall this study was well thought and done. However, estimating migration rates between North and South will help clarify some pending questions.

The sample size of North and South sites combined is good. However, the sample size for individual sites is small. Therefore, conclusions based on comparisons between sites might not be reliable. I would compare individual sites with at least 20 samples. Sites with small sample size should be combined or left out of the analyses.

**Results**

-Does the analysis presented match the analysis plan?

-Are the results clearly and completely presented?

-Are the figures (Tables, Images) of sufficient quality for clarity?

Reviewer #1: yes to all

Reviewer #2: (No Response)

Reviewer #3: The results from individual based analyses (Structure) do not explicitly corroborate the the results from population based analyses (Fst). 

I would suggest one more individual based analyses since your conclusions are based on them. It would be interesting to estimate migration rates between your North and South sites that seems to be differentiated. You could use migrate-N, BayesAss, Geneclass, etc. Then you would have a sense of historical or first and second generation migrants between these sites.

**Conclusions**

-Are the conclusions supported by the data presented?

-Are the limitations of analysis clearly described?

-Do the authors discuss how these data can be helpful to advance our understanding of the topic under study?

-Is public health relevance addressed?

Reviewer #1: yes to all

Reviewer #2: (No Response)

Reviewer #3: Again the results from individual based analyses support the conclusion of limited gene flow across the highway. However, the authors did not estimate migration rates among their sites. 

Figure 2 indicates strong admixture within the sites and does not support limited gene flow.

Overall microsatelite data produces higher Fst estimates when compared to bi-allelic data and the conclusions are based on the Fst estimates and dendrogram. However, you forgot to show the support for each not on figure 2A.

**Editorial and Data Presentation Modifications?**

Reviewer #1: see attached file

Reviewer #2: (No Response)

Reviewer #3: Figure 1A show the supporting values for each node.

**Summary and General Comments**

Reviewer #1: see attached file

Reviewer #2: (No Response)

Reviewer #3: Overall, you can combine sites with small sample size that are near each other. You should also estimate migration rates (historical or recent).

PLOS authors have the option to publish the peer review history of their article (what does this mean?). If published, this will include your full peer review and any attached files.

Reviewer #1: No

Reviewer #2: No

Reviewer #3: No
---

## [Editor Report · Decision Letter 1]

13 Jan 2021

Dear Dr. Watanabe,

The manuscript has been revised following the comments of the reviewrs.

We are pleased to inform you that your manuscript 'The influence of roads on the fine-scale population genetic structure of the dengue vector Aedes aegypti (Linnaeus)' has been provisionally accepted for publication in PLOS Neglected Tropical Diseases.

Best regards,

Mariangela Bonizzoni

Associate Editor

Francis Jiggins

Deputy Editor

---

## [Editor Report · Acceptance letter]

22 Feb 2021

Dear Dr. Watanabe,

We are delighted to inform you that your manuscript, "The influence of roads on the fine-scale population genetic structure of the dengue vector Aedes aegypti (Linnaeus)," has been formally accepted for publication in PLOS Neglected Tropical Diseases.

Best regards,

Shaden Kamhawi

co-Editor-in-Chief

Paul Brindley

co-Editor-in-Chief
